# Caesarean birth in public maternities in Argentina: a formative research study on the views of obstetricians, midwives and trainees

Carla Perrotta [ID] ,[1] Mariana Romero,[2,3] Yanina Sguassero,[4] Cecilia Straw,[5] Celina Gialdini,[4] Natalia Righetti,[3] Ana Pilar Betran,[6] Silvina Ramos[3]

For numbered affiliations see end of article.

**Correspondence to**
Dr Carla Perrotta;
carla.perrotta@ucd.ie

## ABSTRACT

**Objectives** To explore obstetricians', midwives' and trainees' perceptions of caesarean section (CS) determinants in the context of public obstetric care services provision in Argentina. Our hypothesis is that known determinants of CS use may differ in settings with limited access to essential obstetric services.

**Setting** We conducted a formative research study in 19 public maternity hospitals in Argentina. An institutional survey assessed the availability of essential obstetric services. Subsequently, we conducted online surveys and semistructured interviews to assess the opinions of providers on known CS determinants.

**Results** Obstetric services showed an adequate provision of emergency obstetric care but limited services to support women during birth. Midwives, with some exceptions, are not involved during labour. We received 680 surveys from obstetricians, residents and midwives (response rate of 63%) and interviewed 26 key informants. Six out of 10 providers (411, 61%) indicated that the use of CS is associated with *the complexities of our caseload*. Limited pain management access was deemed a potential contributing factor for CS in adolescents and first-time mothers. Providers have conflicting views on the adequacy of training to deal with complex or prolonged labour. Obstetricians with more than 10 years of clinical experience indicated that fear of litigation was also associated with CS. Overall, there is consensus on the need to implement interventions to reduce unnecessary CS.

**Conclusions** Public maternity hospitals in Argentina have made significant improvements in the provision of emergency services. The environment of service provision does not seem to facilitate the physiological process of vaginal birth. Providers acknowledged some of these challenges.

## BACKGROUND

In recent decades, there has been a sustained and unprecedented increase in caesarean section (CS) rates worldwide, in particular—but not exclusively—— in the Latin America region.[1] Argentina, a middle-income country, reported rates between 27% and 52% within the public sector in 2017, while

## Strengths and limitations of this study

► This study is one of the few studies exploring the perceptions of providers working in low-resourced settings in middle-income countries on the determinants of increasing trends of caesarean section (CS).

► Large sample and representation of all professionals and obstetric tasks working in low-resourced settings.

► Cluster analysis allowed researchers to describe the response variability across professional groups in relation to specific CS determinants.

► The use of formative research is a valuable tool to inform the design and implementation of future interventions.

► Even though the response rate was good (63%), those who did not respond may have different views on the determinants of CS.

official data from the private sector is unavailable.[2] According to reports by the Perinatal Reporting System, from 2009 to 2017, the use of CS has increased in the public sector by 22%, from 28% to 34%, with striking rates in some provinces being close to 50%.[2] Similarly, Brazil registered national rates of 55%, with private providers close to 90%.[3]

The determination of the ideal CS rate remains controversial. A WHO systematic review of ecological studies concluded that CS rates above the threshold of 9%–16% were not associated with decreases in maternal and infant mortality.[4] Another global analysis led by the WHO also concluded that once the CS rate reaches 10%, further increases had no impact on maternal, neonatal and infant mortality rates. Although the exact ideal CS rate is unknown, the WHO states that at population level, CS rates higher than 10% are not associated with reductions in maternal and newborn mortality rates; thus, current regional trends in Latin America

cause concern and call for a scaling-up of interventions to reduce unnecessary CS.[5 6]

There are multiple factors underpinning the increase in CS rates.[7] Notably, financial incentives, the organisation of obstetric services, obstetricians' fear of litigation and professional attitudes towards CS.[8–13] Systematic reviews show that, contrary to perceived opinion, women prefer to give birth vaginally.[14 15] According to a recent scoping review, women's preferences for CS are in the range of 5%–20% depending on the female population characteristics, that is, age, ethnicity, cultural background, social and economic status, level of education and at what stage in the pregnancy that question is asked or how the question of preference is framed.[15]

Along with clinical interventions, non-clinical interventions such as antenatal education, continuous emotional support during labour, audit and feedback, and the implementation of clinical guidelines have shown modest but statistically significant reductions in CS rates.[7 16 17] A quality improvement study conducted in Brazil presented encouraging results. The project demonstrated a reduction in CS rates through a multilevel intervention targeting healthcare professionals. The interventions included reinforcing the use of analgesia during birth, continuous professional education and feedback over CS rates to the teams, among other measures.[18] The WHO also recommends implementing non-clinical interventions to reduce unnecessary surgical births and it encourages the incorporation of an initial formative research process to tailor interventions to local contexts, ensure implementation feasibility and anticipate specific barriers or identify facilitators.[19 20]

Argentina is a federal country with a two-tier health system: a public system financed by government funds and a private system mix of social insurance and private insurance. Virtually all births occur at healthcare institutions (99.3%), and 64% of them within the public health system. Specialists lead obstetric services in obstetrics and gynaecology (OB/GYN), while the role of midwives depends on how the hospital organises its services and human resources. Midwives provide antenatal care and, depending on the institution, can assist or lead low-risk vaginal births.

We conducted formative research to enrich understanding of the reasons for the sustained increase in CS use in Argentina's public sector with the aim to subsequently use this knowledge to tailor interventions in Argentina.[21] We applied a mixed-methods approach to explore multiple layers and perspectives of both women and health providers and to assess the characteristics of the obstetric services in the participating hospitals including the availability of and access to pain management interventions (eg, anaesthesia and non-pharmacological interventions) and the availability of and access to emergency obstetric services. Formative research has its origins in ethnographic studies and uses various data-gathering processes. The rationale behind this approach is to consider the peculiarities of the setting to better tailor

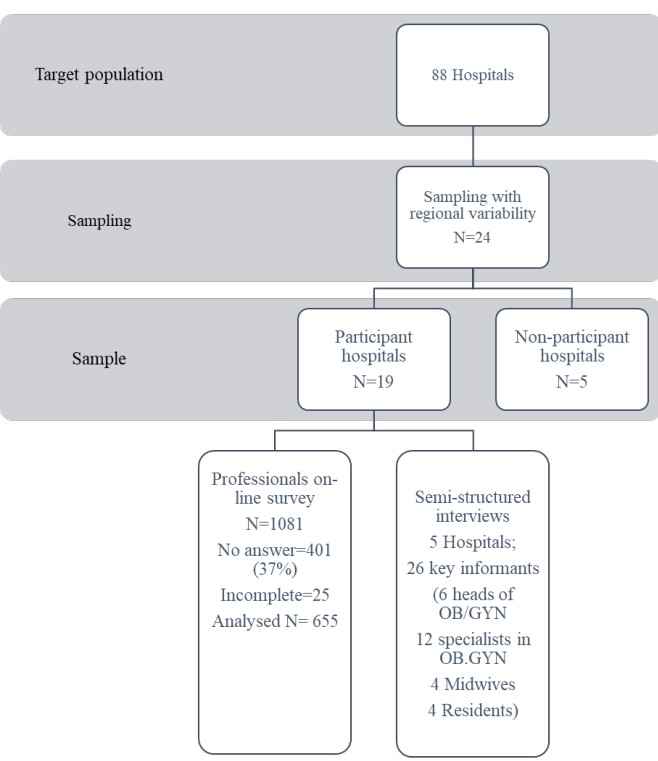

**Figure 1** Flow chart of the hospital selection for the on-line survey component and semistructured interviews. OB/GYN, obstetrics and gynaecology.

suitable interventions and, at the same time, to catalyse the target population into change.[22–24]

This article will focus on the perspectives of specialists in OB/GYN, midwives and OB/GYN residents (trainees) on determinants of CS use in Argentina's public maternity hospitals.

## METHODS

A comprehensive description of the methodology has been published elsewhere.[21] Briefly, we used a research approach that combined quantitative and qualitative data gathering techniques conducted by a multidisciplinary team.

The contextual factors of the study refer to the adequacy of services to provide emergency obstetric care, human resources availability and women's access to comprehensive obstetric care like antenatal education, companionship during birth and holistic pain management.

The eligibility criteria for hospitals to participate in the study were: (A) public hospitals with ≥1000 annual childbirths and (B) use of the National Perinatal Information System.[25 26] Out of the 88 potentially eligible institutions, 24 were non-randomly selected to represent the six Argentinian regions. Nineteen agreed to participate and obtained approval from local Ethics Committees (figure 1), and five institutions declined to participate. Finally, five Argentinian regions were included in this

study: Centre, North, Northeast and Buenos Aires City. In 2016, the median CS rate in the 24 eligible hospitals was 37%, ranging between 27% and 52%.[2]

The report of this research follows the Standards for Reporting Qualitative Research and Formative Research and the Strengthening the Reporting of Observational Studies in Epidemiology cross-sectional checklist.[27–29]

## Subjects and data collection

Multiple data sources were used: an institutional survey, a healthcare provider online survey and semistructured interviews with key informants. The head of OB/GYN identified a spokesperson who led the fieldwork and introduced the project to health team members at each participating hospital.

### Institutional survey

The selected champion in each hospital completed the institutional survey. The survey gathered information on the number of live births in 2017, the number and type of providers involved in maternal care, hospital CS rates, institutional infrastructures such as the number of beds, availability of services to safely manage obstetric emergencies, availability of antenatal education for women and pain management during labour and birth.

### Online survey for healthcare providers

Eligible participants for the online survey included all specialised physicians, obstetricians and residents at OB/GYN services at participating hospitals. Both permanent staff and shift personnel were included.

The health professionals' online survey included participation consent (mandatory answer) and 32 questions: 26 with pre-established options and 6 open-ended questions. The survey gathered the following data: role and healthcare work, including the number of deliveries/CSs performed in a week; perception of determining factors for the use of CS related to the institution, health professionals and pregnant women; frequency of use of monitoring cycles; improvement in quality of care; and Robson Classification knowledge, among others.[30]

The piloting process involved 10 hospitals and asked participants to indicate if they would like to add anything. Fifteen providers suggested that inadequate pain management during labour is a contributing factor; thus, we incorporated it in the list of determinants of CS use ('*lack of pain management interventions is associated with the use of CS*').

We sent an email with the link to the survey to each eligible subject who provided an email address; the text included a brief presentation of the study with a consent form. The respondents' gender or sex was not recorded to ensure anonymity. There were 3 weekly reminders. Fieldwork was carried out from July 2018 to August 2019.

### Semistructured interviews for key informants

Out of the 19 hospitals participating in the study, we drew a convenience sample of five hospitals representing the different regions. In those hospitals, we conducted in-depth interviews after the online survey with key informants, including the heads of departments or services, specialists in OB/GYN, midwives and residents. The interviewers were researchers with experience in qualitative methods and in maternal health and organisational culture. The interviews were conducted either in person or by video call and recorded under informed consent.

### Data analysis

We described variables using standard statistical parameters. Continuous variables were presented as means and SDs, whereas categorical variables were presented as numbers and percentages. Human resources data were standardised to 1000 live births in 2017.

For the online survey, answers were grouped into three categories: disagreement (strongly disagree and disagree), neutral and agreement (completely agree and agree) and were described using frequencies. The statements with modest consensus—those with ≤50% agreement or disagreement—were analysed using cluster analysis to examine if professional groups or any institutional variable explained the variance. Given the different types of variables assessed, a Gower's distance matrix was calculated. It was visualised using the t-Distributed Stochastic Neighbour Embedding (t-SNE, 1) and clustered with partition around medoids with k=4.[31 32] All statistical analyses were performed using R V.3.6.3.[33]

We transcribed the recorded in-depth interviews. Transcripts were independently coded by two researchers with experience in qualitative analysis. We undertook an in-depth framework analysis of the qualitative data focusing on determinants of CS and institutional factors.[34] We compared key themes emerging from the interviews with the online survey results, expanding to supplement survey findings when appropriate. Throughout the results section, we report verbatim extracts to validate and illustrate specific issues, provide further explanations or include themes that were not addressed by the online survey.

### Patient and public involvement

The development of the protocol used evidence from qualitative research exploring women's experiences with obstetric services in public maternity hospitals. Once the study has been published, the results will be disseminated for professional, non-professional audiences in participant provinces, as well as to non-governmental organisations overseeing maternal care in the Latin American region.

## RESULTS

Women giving birth in the 19 participant hospitals represented 19% of the total live births in Argentina's public sector in 2017 (figure 1). The hospitals corresponded to the North region (four hospitals), the Northeast (two hospitals), the Centre (seven hospitals) and capital city (six hospitals). Five hospitals (three from the West-Cuyo

Region and two from the south) either declined to participate or did not complete the endorsement by their ethics committee.

Table 1 shows the characteristics of the 19 hospitals in 2017 as reported by the key informants in each institution. The percentage of live births by CS in the study hospitals ranged from 29% to 40% (table 1). The hospitals had similar services; all of them had laboratory, neonatal intensive care units, adequate access to emergency CS and resources to manage obstetric emergencies, including safe blood transfusion services. All hospitals but one had limited access to pain management during birth, insufficient provision of antenatal education and restricted companionship during CS. They differed on the number of healthcare providers per 1000 live births, with a median of seven obstetricians/1000 live births and 2·8 midwives/1000 live births. Two hospitals reported a very low staffing ratio—one hospital reported one obstetrician/1000 live births—and the largest number of obstetricians was 23/1000 live births in one hospital. All the professionals received a fixed monthly salary unrelated to performance.

We received 680 online surveys (response rate of 63%; range 19%–84%). We excluded 25 due to missing data related to professional background, which resulted in 655 completed surveys to analyse (figure 1).

The completed surveys were from OB/GYN specialists (n=327), midwives (n=161) and residents (n=167). Cluster analysis allowed us to identify four groups based on tasks and years of experience, and profession. Group 1 (n=247) included OB/GYNs who worked for a mean of 11 years and attended 2–10 births per week (any type). Group 2 had more experienced OB/GYNs (n=80) and midwives (n=23) (mean 22 years of professional experience) involved either in antenatal care, teaching or management duties. Group 3 consisted of residents (n=167) with 3 years of experience participating in more than 10 deliveries per week. Group 4 included midwives (n=138) with a mean professional practice of 12.5 years in assisting vaginal deliveries (table 2). We conducted 26 semistructured interviews in total with six heads of OB/GYN departments, 12 specialists in OB/GYN, 4 midwives and 4 residents representing each of the regions in this sample (North: 2 hospitals, East: 1 hospital, Centre: 1 hospital and Buenos Aires City: 1 hospital).

Six out of 10 participants (411, 61%) agreed that the use of CS is associated with '*the complexities of our caseload*' (table 3). Additionally, key informants point out diagnostic innovations and comorbidities as determinants of CS use.

> Nowadays, we have more diagnostic procedures during pregnancy […] Maybe, in the past, women had a vaginal birth with worse clinical outcomes […] yes, the number of CS has increased, but the neonatal outcomes have improved. (Hospital 2, OB/GYN)

> The main factor for CS is clinical. We see more women with severe comorbidities. Also, if they have had

**Table 1** Characteristics of participant hospitals (n=19)

| Variables | Hospitals N (%), mean (SD), median (range)† |
|---|---|
| Type of institution | |
| Maternity | 7 (37%) |
| General hospital | 12 (63%) |
| Number of beds * | |
| Obstetrics | 44 (1–93) |
| Neonatal intensive care unit | 18 (1–50) |
| Number of delivery rooms or units of labour, delivery and recovery* | 6 (1–16) |
| Number of live births | 52 633 2355 (1025–9633)* |
| Caesarean sections (CS) | |
| Number of CS | 21 102 |
| CS rate* | 37.2 (27–50) |
| Protocol for vaginal delivery after CS | 15 (83%) |
| Residency programme | |
| OB/GYNs | 18 (90%) |
| Midwives | 7 (36%) |
| Number of professionals* | |
| OB/GYNs | 20 (2–67) |
| Residents* | 13 (0–31) |
| Midwives* | 12 (0–30) |
| Number of professionals/1000 live births* | |
| OB/GYNs | 7 (1–23) |
| Residents | 5.8 (0–15) |
| Midwives | 2.8 (0–18) |
| Midwives' availability (24/7) | 13 (68%) |
| Midwives participating in low-risk births | 9 (45%) |
| Pain management interventions | |
| Access to non-opioids | 10 (52%) |
| Access to epidural (24/7) | 9 (47.4%) |
| Access to opioids | 12 (63%) |
| Relaxation techniques | 4 (21.1%) |
| Access to hot shower | 9 (47.4%) |
| Access to massages | 3 (15.2%) |
| Access to companionship (24/7) | |
| During labour | 14 (73%) |
| During delivery | 16 (84%) |
| During CS | 2 (10%) |
| Availability of prenatal education (morning, afternoon, Monday–Friday) | 1 (5.5%) |

*Median (range).
†Data are shown as N (%) unless otherwise indicated.
OB/GYN, Specialist in Obstetrics and Gynaecology.

**Table 2** Online survey clusters according to clinical experience (years) and main duties

| | Median years of professional experience | Main duties | n=655 |
|---|---|---|---|
| Group 1 OB/GYNs | 11 | Assistance of all delivery modes. 2–10 deliveries per week | 247 |
| Group 2 OB/GYNs and midwives | 22 | Antenatal care, research, supervisory tasks, education | 103 |
| Group 3 Residents | 3.5 | Assistance of all delivery modes. More than 10 deliveries per week | 167 |
| Group 4 Midwifes | 12.5 | Assistance of all deliveries 2–10 deliveries per week | 138 |

OB/GYN, obstetrics and gynaecology.

a previous CS and comorbidity, the delivery is a CS […]. (Hospital 3, OB/GYN)

Shortage of skilled professionals was considered a determinant of the overuse of CS by 12.8% of the sample, while 16% agreed that insufficient training for complicated births was a determinant.

A similar level of agreement appeared regarding inadequate skills to conduct vaginal birth after CS (n=84, 12.4%) (table 3). However, when asked if CS is associated with residents' lack of training in performing complex deliveries, the professional groups differed in their answers: 30% of midwives agreed that there is an association (n=39, 30%), while only 5% of residents agreed (table 4).

The semistructured interviews allowed us to explore further whether the number, skills and experience of obstetricians, midwives and residents influence the use of CS. Among the interviewees, there was consensus that professionals are well trained; however, midwives perceived themselves as less prone to intervene and considered that the medical professionals were trained with a more interventionist mindset.

We have to fight every day as doctors tend to intervene. Now we have a training program on foetal medicine, and most doctors perform a foetal ultrasound. I notice more and more the tendency to intervene. (Hospital 1, midwife)

The semistructured interviews also revealed challenges in training future generations of obstetricians to perform instrumental deliveries when needed.

**Table 3** Online survey answers according to levels of agreement (n=655)

| | | Disagree or strongly disagree N (%) | Agree or strongly agree* N (%) | Neutral N (%) |
|---|---|---|---|---|
| | **In my institution, the use of CS is associated with…** | | | |
| Statements with higher % of high and moderate agreement | The complexity of our case load | 138 (20.4) | 411 (60.8) | 112 (16.5) |
| | Financial incentives | 621 (91.8) | 14[2] | 24 (3.5) |
| | CS being safer than a VB | 549 (81.2) | 40 (5.9) | 71 (10.5) |
| | Shortage of human resources | 522 (77.2) | 86 (12.7) | 54 (7.9) |
| | Deficits in infrastructure | 419 (73.8) | 89 (13.1) | 71 (10.5) |
| | Inadequate number of trained healthcare professionals with the skills to perform complex deliveries | 464 (68.6) | 87 (12.8) | 98 (14.5) |
| | Deficit in training to conduct VB after a CS | 467 (69) | 84 (12.4) | 106 (15.6) |
| | Lack of access to pain management strategies during VB† | 163 (44) | 116 (31) | 85 (12.7) |
| Statements with higher % of neutral answers or low level of agreement | Deficits in the training of residents to monitor and perform complex deliveries | 418 (61.8) | 104 (15.3) | 138 (20) |
| | Fear of litigation | 239 (39.1) | 225 (36.8) | 147 (24) |
| | First time mothers and their anxiety and poor preparation for VB | 246 (37.4) | 221 (33.7) | 188 (28) |
| | Women's preferences | 290 (44.2) | 172 (26.4) | 193 (29.3) |

*High and moderate agreement: more than 60% agree or disagree.
†This statement was incorporated in the second round of surveys.
CS, caesarean section; VB, vaginal birth.

**Table 4** Online survey responses with the lowest level of agreement in clusters grouped by profession, years of experience and clinical tasks*

| | Cluster 1 Group 1 OB/GYN specialists 11 years of experience n=247 | Group 2 OB/GYNs and midwives 22 years of experience in antenatal care, research and supervisory tasks n=103 | Group 3 Residents n=167 | Group 4 Midwives n=138 |
|---|---|---|---|---|
| 'I agree or completely agree that the use of CS is associated with deficits in the training of residents to monitor and perform complex deliveries' | 40 (16) | 18 (17) | 8 (5) | 39 (30) |
| 'I agree or completely agree that the use of CS in first time mothers is related to their anxiety as they don't know birth can be a long process' | 94 (38) | 30 (28) | 57 (34) | 40 (29) |
| 'I agree or completely agree that the use of CS is associated with women's preferences' | 74 (30) | 26 (27) | 48 (28.6) | 24 (17) |

*Cluster analysis Gower distance matrix partition around medoids.
CS, caesarean section; OB/GYN, obstetrics and gynaecology.

Instrumental deliveries are tough, and no one wants to do them. Before, it was a common practice; now, people are cautious. In the collective imagination of society, the forceps can kill or harm the baby. Even judges, lawyers have that idea… [And the training to conduct instrumental deliveries?] … little as we don't do them. (Hospital 4, OB/GYN)

A resident can complete their training—if lucky—performing five forceps. Maybe a handful of obstetricians at the ward have the skills to use forceps. Doctors prefer a CS as it can quickly solve the clinical problem. (Hospital 2, Resident)

In most hospitals, key informants did not support the idea that scarcity of human resources is relevant to the increase of CS. However, in two hospitals with either no midwives on their teams or a very low staff-to-live-birth ratio, clinicians did recognise it as a contributing factor. In the latter two hospitals, busy shifts with only one doctor on-call were described. In those circumstances, if a woman presented with failing-to-progress labour, the decision may switch to a surgical delivery (verbatim not shown).

Questions related to women's preferences and first-time mothers' anxiety due to lack of preparation for birth had the highest number of neutral responses. We found variability between professional groups. Seventeen per cent of midwives agreed with the statement '*women demand a CS*' compared with 30% of obstetricians and 28% of residents. For the statement '*the use of CS in first-time mothers is related to anxiety*', 38% of the obstetricians, 34.4% of residents and 29% of midwives agreed it could be considered a determinant of CS use (table 4).

Key informants discussed both matters:

We have requests both ways. Patients that want vaginal birth at all cost and patients that want CS. We work

under significant pressure. Patients have changed… (Hospital 4, OB/GYN)

Lack of adequate access to pain management interventions as a reason for CS did not reach consensus in the online survey; 31% agreed, 44% disagreed and 12.7% had a neutral opinion, with statistically insignificant variation between responses (table 3). Key informants helped to understand the role of limited access to epidurals and other pain management strategies like hot water, relaxation or massage.

There was some consensus that the absence of these options might influence women's requests for a CS, especially in adolescent mothers.

If a patient is in pain, I won't specifically say 'OK, this patient needs a CS because of the pain'. It does not affect our decision (doctors), but it does affect the patient's […] The vast majority of patients ask for a CS because of the pain. (Hospital 4, OB/GYN)

When asked if the lack of epidural analgesia is a determinant of CS use, a midwife in a hospital with no access to an epidural and a high proportion of adolescents eloquently stated:

Totally. I agree. We do not have a population of women aged 30 or more. They are empowered, and they can say: 'Yes, I would prefer a vaginal birth even though it means I will have pain during birth'. [Our population] are younger girls, and to be honest, a fifteen-year-old girl… it is tough to accept labour pain. We do not have access to an epidural for them to at least try a vaginal birth. In those cases, yes, we would recommend a CS. (Hospital 3, Midwife).

Others disagree: '*I don't think pain is per se a limitation during birth. We do have some strategies for pain management,*

*especially midwives do. They know how to apply and use them. Thankfully, women can choose their companion during birth, and that is a big relief* (Hospital 2, OB/GYN).

In a hospital with access to an epidural, the key informants did not see birth pain as a CS determinant:

We did not notice that pain [affects decision-making]. We do try to use epidurals, especially with younger girls […] But, no, not at all, I don't think pain influences CS. (Hospital 1, Midwife)

Fear of litigation was relevant for the two professional groups with more experience—OB/GYN and midwives performing more than 10 deliveries per week, group 1 (n=151, 73%) and OB/GYNs and midwives currently in teaching, management or research tasks, group 2 (n=67, 65%) (results not shown)—while 40% and 37% of overall online survey responses disagreed and agreed, respectively (table 3).

Unfortunately, as obstetricians, we are the most litigated specialists in the country. It's complex. (Hospital 2, OB/GYN)

We have learnt that we are alone. Professional liability insurance does not cover us [in the case of litigation]. We must respond with our savings. Maybe with all your assets. We have learned that the lawyers defend the hospital in public hospitals, but not us, the professionals working in the hospital. You would think that they would defend the doctors, but they don't […]. (Hospital 1, Specialist).

There was consensus among all professional groups that the use of CS in public maternity hospitals is unrelated to financial incentives for the institution (n=621, 92%), the perception that CS is safer than vaginal birth (n=549, 81%) or unsupportive infrastructure (n=419, 74%) (table 3).

Most of the participants did not recognise any formal communication of CS rates by their institutions. About half of the participants (n=380, 56·2%) expressed having regular meetings to audit clinical indications of CS. A quarter indicated that audit activities were in response to an unexpected adverse event rather than routine (n=161, 23%). Finally, 6 out of 10 healthcare professionals agreed that there was a need to optimise CS use and half indicated they were familiar with the Robson classification with no differences in terms of their opinion on the influence of clinical caseload and the knowledge of the Robson classification (data not shown).

## DISCUSSION

This formative research study characterises public maternity hospitals' work environment in Argentina, emphasising the availability of essential obstetric services and human resources. It provides an improved understanding of healthcare providers' perceptions of determinants of CS in the public health system and a framework for anticipating which non-clinical interventions are feasible and valid considering the factors mentioned previously.

Our research suggests that services are organised to provide adequate responses to obstetric emergencies. However, they informed deficits in the availability of strategies for pain management such as epidural, hot showers or relaxation techniques, and limited antenatal education, all of which are important issues for women.[35] Some hospitals had insufficient human resources and space limitations impeding constant support during birth. Evidence worldwide suggests fear of pain or lack of pain relief during labour as one of the primary factors underlying women preference for CS.[7 15 36 37] Our formative research further reinforces the importance for women to have access to pain management strategies. The findings indicate that women from these hospitals prefer vaginal birth over CS, but CS preference is influenced by the pain experienced during birth.[38]

The interviews with key informants acknowledged restrictive access to pain relief and its impact on the birth experience, particularly in adolescents. The management of pain during birth is an essential component of obstetric services and needs to be organised, resourced and monitored. This is particularly true in Argentina, which has a 13% overall adolescent pregnancy rate and provinces with rates above 20%.[2] Moreover, a recurring topic that appeared when conducting semistructured interviews was the need to improve antenatal education, which has already been identified as a non-clinical intervention with a positive impact on birth experience and vaginal birth rates.[7 35 39]

All the professional groups agreed that CS is related to complex-case mix or clinical indications. This finding is in line with results from other international studies. Panda *et al*[11] summarised clinicians' views of factors influencing decision making in a systematic review of 34 studies published in English and clinical indication appeared significant in eighteen studies. Still, the reported CS rates by most of the hospitals in this sample—median of 37%—suggest that case-mix alone cannot explain the high rates. Our online survey did not explore what providers understood as a clinical indication, but the semistructured interviews further confirmed clinicians' views that increased comorbidities and new technologies contribute to the increased use of CS. These findings suggest that overall, healthcare teams perceived clinical indications as the main driver for performing a CS though about a third identified other contextual factors as determinants of CS use like services organisation, service provision, limited human resources, midwives' role, women's preferences, adequate number of professionals to support women during labour or deficits in antenatal education for both pregnant women and companions. Though factors such an increase in the age of women, in the proportion of nulliparous women, in the prevalence of obesity or in multiple births (in vitro fertilisation) have been suggested as culprits of the increasing trend of the use of CS, these factors alone cannot explain the phenomenon. The

WHO emphasises that non-clinical factors such as convenience of a CS, fear of litigation and other organisational issues need to be recognised and addressed for successful interventions.[39]

In addition, the WHO recommends the use of the Robson classification to assess, monitor and compare CS rates in a standardised and action-oriented manner.[19 30] This classification helps to understand the obstetric case mixed of a maternity and thus can challenge some common myths about causes of increasing CS rate such as obstetrical condition conducive to CS. In our survey, only about half of the healthcare providers had knowledge of the Robson classification. Interventions to increase knowledge and use of the Robson classification could be beneficial in these maternities. Since our study was not designed to analyse how the knowledge of the Robson classification may influence clinicians' opinion on the use of CS or the current rates, it is not possible to make inferences on the association between these two variables.

From the healthcare teams' perspective, the implementation of interventions targeted at clinicians like audit and feedback, clinical guidelines and the introduction of a second opinion may help to improve their clinical decision making. Still, this study showed the importance of assessing contextual factors such as services provision and human resources. If clinicians have limited options for pain management or no human resources to provide continuous labour support—in some institutions without midwives—the interventions focusing on clinical guidelines or audit and feedback alone may not lead to a desirable result.

Residents' skills emerged as a controversial theme in our study. Most residents perceived that their training to perform difficult vaginal births is not related to the increased use of CS, but more than half of midwives disagreed. As reported in many other countries, our data suggest that the use of instrumental deliveries is in decline, with consequently reduced possibilities for training new obstetricians. How to better equip trainees for instrumental birth requires further research.[40 41]

We find differences in health providers' views of the role of women's preferences for CS. Midwives actively involved in antenatal care or assisting deliveries disagreed with the view that women's preferences are linked to CS use, which is also in line with previous research worldwide. However, in half of the participating hospitals, midwives are not present to assist during birth. The power imbalance between medics and midwives is well documented. It can be addressed by changes in human resource organisation or by providing midwives with an active role on the team. Our study did not explore power imbalance, but it showed that obstetricians lead labour in most hospitals, which results in a very limited role for midwives during birth, even in low-risk pregnancies.[8 16] Once again, our study showed that the small number of midwives in most of the hospitals is insufficient to guarantee continuous professional support during birth for women. Considering the limited exposure to antenatal

education in addition to this, the result is an environment that does not seem to facilitate the physiological process of vaginal birth.

In terms of insufficient infrastructure or trained human resources, we mistakenly assumed that providers would acknowledge these factors as contributing to the overuse of CS. However, only providers in hospitals with severe shortage of human resources recognised the influence on clinical decision making.

The extent to which fear of litigation impacted obstetricians with more clinical experience requires all stakeholders to find possible solutions. There are extensive references to lawsuits against obstetricians in the literature and evidence. We look at this potential determinant of CS as a burning issue in public maternity hospitals in Argentina. The fear of litigation was deeply rooted among experienced obstetricians in public hospitals as they feel they work in an environment with minimum support. The road to sort out litigation in healthcare is complex compared with the experience of high-income countries but needs to be addressed if CS use is to be reduced in a sustainable manner.[42–44] The disagreement regarding the association between financial incentives and surgical deliveries is not surprising. This association is relevant for physicians paid in a fee-for-practice model. In the public sector of Argentina, healthcare providers receive a fixed monthly salary regardless of the workload or the procedures they perform, and hospitals do not usually receive extra funding for a surgical delivery.

## Strengths and limitations

To our knowledge, our study is the first to explore the perceptions of providers working in public hospitals in Argentina on the determinants of increasing trends of CS and future interventions to optimise its use. We showed that the use of formative research is a valuable tool to inform the design and implementation of future interventions. We had a large sample and representation of all professionals and obstetric tasks, which permitted us to identify differences across professional cadres. This variability allows for tailoring implementations specifically to each cadre, which will also benefit the implementation process. Participants' response rate was high considering the usual response rates from providers. However, professionals who did not complete the survey may not engage with the implementation process and therefore result in an additional barrier.

Another limitation is that opinions do not necessarily reflect current behaviour. Nevertheless, perceptions are an essential component of how health teams frame their environment and give sense to their clinical decisions.

## CONCLUSIONS

There is a consensus among obstetricians, midwives and residents on the need to implement interventions to decrease unnecessary surgical deliveries in Argentina.

We highlight an essential finding: obstetric services need to incorporate a holistic approach to pain management during vaginal birth and ensure continuous support during delivery from staff or carers to optimise CS use, especially in adolescents and first-time mothers.

At the same time, providers agreed that clinical indication is the main factor driving CS use and, therefore, interventions supporting the decision-making process can be of benefit.

Fear of litigation emerged as a critical issue highlighted by more experienced obstetricians. Therefore, strategies to protect health providers adhering to obstetric evidence-based guidelines against any legal matter within the public health system are warranted.

**Author affiliations**
[1]School of Public Health, Physiotherapy and Sport Sciences, University College Dublin, Dublin, Ireland
[2]Consejo Nacional de Investigaciones Científicas y Tecnológicas (CONICET), Buenos Aires, Argentina
[3]Health, Economy and Society Department, CEDES, Buenos Aires, Argentina
[4]Centro Rosarino de Estudios Perinatales, Rosario, Argentina
[5]School of Social Sciences, University of Buenos Aires, CEDES, Buenos Aires, Argentina
[6]UNDP-UNFPA-UNICEF-WHO-World Bank Special Programme of Research, Development and Research Training in Human Reproduction (HRP), Department of Sexual and Reproductive Health and Research, World Health Organization, Geneve, Switzerland

**Acknowledgements** Dr Marcelo Soria, Associate Professor, University of Buenos Aires, provided valuable statistical advice. Mercedes Vila Ortiz edited the manuscript. Carolyn Ingram edited the revised version of the manuscript.

**Contributors** All authors participated in developing this project protocol. SR was responsible for the scientific aspects of the project, coordinating the team, protocol development, as well as for writing and reviewing the final version of the original protocol. APB was responsible for the revision of the protocol and for technical assistance regarding the design and methodology. She also obtained the funding for the project. MR, CP and YS were responsible for assisting the writing of the protocol and developing the versions of the instruments. NR and CG contributed to the development of the instruments and coordinated field work. CP, MR and YS contributed to the writing of the article and reviewed the final version. All authors provided feedback and revised the manuscript. The following authors were responsible for specific components of the project: SR wrote the first version of the study protocol and, together with MR, CP and YS, coordinated its development and approved the final version. CP contributed to the design of the methodology and analysis of the project. MR contributed to the methodology, the design of the fieldwork. YS contributed to literature revision. YS, CS, CG and NR conducted the semistructured interviews. CS and MR conducted the semistructured interviews analysis. CP is the guarantor of this research work. All authors read and approved the final manuscript.

**Funding** This research was funded by the UNDP-UNFPA-UNICEF-WHO-World Bank Special Programme of Research, Development and Research Training in Human Reproduction (HRP), a cosponsored programme executed by the WHO in the Department of Sexual and Reproductive Health and Research. WHO Project Number A65919. CP also received funds to complete this research from the University College Dublin, College of Health Sciences, SEED Fund (2019) Grant number SF1735. NR is a research fellow funded by the Ministry of Health, Buenos Aires City Government.

**Disclaimer** The study design; the collection, analysis and interpretation of the data; the writing of the report and the decision to submit the paper for publication are solely the responsibility of the authors and do not reflect the views of the referred programprogramme or the other funding institutions. All authors had full access to all the data and accept responsibility to submit for publication.

**Competing interests** None declared.

**Patient and public involvement** Patients and/or the public were involved in the design, or conduct, or reporting, or dissemination plans of this research. Refer to the Methods section for further details.

**Patient consent for publication** Not applicable.

**Ethics approval** The project was approved by the Independent Ethics Committee of Centro Rosario de Estudios Perinatales and by the provincial Ethics Committees and/or the Teaching and Research Committees at each of the selected hospitals pursuant to the requirements in each jurisdiction. It was also approved by the Research Project Review Panel of the UNDP/UNFPA/UNICEF/WHO/World Bank Special Programme of Research, Development and Research Training in Human Reproduction at the Department of Sexual and Reproductive Health and Research of WHO, and the WHO Research Ethics Review Committee, Geneva, Switzerland. Reference Number A65919. In Argentina, the research protocol was registered in the RENIS database (number IS002316). Participants gave informed consent to participate in the study before taking part.

**Provenance and peer review** Not commissioned; externally peer reviewed.

**Data availability statement** Data are available on reasonable request. The data will be stored on CEDES' server, encrypted. CEDES will be the guardian of the dataset. Dataset is available on reasonable request. Data are anonymised. Our data collection forms do not include any variable that could reveal the identity of the participants or that potentially could identify the participant institutions.

**ORCID iD**
Carla Perrotta http://orcid.org/0000-0001-5986-4581

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
