## [Reviewer comments · BMJ Open]

ARTICLE DETAILS

TITLE (PROVISIONAL)	Caesarean birth in public maternities in Argentina: a formative research study on the views of obstetricians, midwives, and trainees.
AUTHORS	Perrotta, Carla; Romero, Mariana; Sguassero, Yanina; Straw, Cecilia; Gialdini, Celina; Righetti, Natalia; Betran, Ana Pilar; Ramos, Silvina

VERSION 1 – REVIEW

REVIEWER	Negrini, Romulo Santa Casa São Paulo - Brazil, Obstetrics and Gynecology
REVIEW RETURNED	29-Jun-2021

GENERAL COMMENTS	This is an interesting article that shows the hospitals staff's view of the high C-section rate. It brings also, the ability of latin countries produces good researches with limited budgets. Despite this I would like to propose some improvements in order to make the article better to serve as reference to orders researches, and to clarify some doubts. 1) in the introduction the authors say that the C-section rate in Argentina is high, but I think is necessary to specify the ideal rate according some health society or organization, for example, WHO. 2) in the introduction also, in the second paragraph, the authors discuss about global initiatives to reduce C-section rates. But I missed the article that have been published recently in BMJ open quality describing a successful initiative in a public Brazilian hospital to reduce the local C-section rate (Negrini R, Ferreira RDdS, Albino RS, et al. Reducing caesarean rates in a public maternity hospital by implementing a plan of action: a quality improvement report. BMJ Open Quality 2020;9:e000791. doi:10.1136/ bmjoq-2019-00079). 3) the authors cited, in the introduction also, that the c-section rate has been increasing in Argentina but they did not bring the temporal evolution of this rate. I think is important to the readers to have this information. 4) Why the C-section rate of 27% was chosen by the authors as cutoff for choosing the study hospitals? 5) It's clear that hospitals of all Argentina's regions was chosen, but I think is important to specify how many hospital from each region. 6) the absent of Pain relief is a known issue the lead pregnant to choose the c-section as a delivery mode. I think it could be more discussed. It's interesting to bring the pregnant perspective raised in other articles, for exemple, Stoll K, Hauck YL, Downe S te al. Preference for cesarean section in young nulligravid women in eight OECD countries and implications for reproductive health
---

	education. Reprod Health. 2017 Sep 12;14(1):116. doi: 10.1186/s12978-017-0354-x. 7) Most of the interviewee believe that clinical conditions are responsible for the increasing in the c-section rate. I guess it is a common excuse for the most of midwives and doctors in the obstetric area. Because of this, Robson created his classification. The authors cited that the knowledge of interviewees regarding Robson classification was part of the survey, but they did not mentioned the results of this question. It would be interesting to build a parallel among these information (Knowledge of Robson's classification x the belief that clinical conditions are responsible for high C-section rates).
--	--

REVIEWER	Litorp, Helena Uppsala Univ
REVIEW RETURNED	20-Oct-2021

GENERAL COMMENTS	Thank you for asking me to review this interesting paper on determinants of CS (over)use in Argentina. I think the paper adds further understanding and knowledge to the increase in CS rates observed globally and specifically in Latin America and with results in line with other reports on the subject. Please find some specific comments below. Introduction  * The introduction is well-structured and interesting, in particular I like the description of formative research and that the authors argue for why they chose this method for the current study. * I would, however, recommend to also state the national CS % in Argentina, preferably both in total, in private and in public sector (that is on the population level). * I would recommend to rephrase the sentence of the aim of the study to include more of the study objectives (which I understand also included a survey on EmOC services available). Methods  * Pls motivate why only public hospitals were included and why hospitals needed to have a CS rate above 27%. Discussion The Discussion is overall well-balanced, but I think the section on obstetric care providers' view that the CS rate is caused by a complex case-mix would benefit from a discussion whether such arguments are warranted - can one really argue that around 40% of deliveries are so complex that they require CS given that complications demanding CS are in fact relatively rare events? Or would this rather be a way to withdraw from responsibility? Pls elaborate.
---

VERSION 1 – AUTHOR RESPONSE

Reviewer 1

Dr. Romulo Negrini, Santa Casa São Paulo - Brazil, Hospital Israelita Albert Einstein

Comments to the Author:

This is an interesting article that shows the hospitals staff's view of the high C-section rate. It brings also, the ability of countries produces good researches with limited budgets. Despite this I would like to propose some improvements in order to make the article better to serve as reference to orders researches, and to clarify some doubts.

1. In the introduction the authors say that the C-section rate in Argentina is high, but I think is necessary to specify the ideal rate according some health society or organization, for example, WHO. Our response:

Thanks for your positive feedback and your suggestion to include the optimal CS rate by WHO or relevant bodies. We have included one paragraph clarifying the current recommendations as follows. The text now reads

“The determination of the ideal CS rate remains controversial. A WHO systematic review of ecological studies concluded that CS rates above the threshold of 9-16% were not associated with decreases in maternal and infant mortality (4). Another global analysis led by the WHO also concluded that once the CS rate reaches 10%, further increases had no impact on maternal, neonatal, and infant mortality rates. Although the exact ideal CS rate is unknown, the WHO states that at population level, CS rates higher than 10% are not associated with reductions in maternal and newborn mortality rates - thus, current regional trends in Latin America cause concern and call for a scaling-up of interventions to reduce unnecessary CS (5,6)”.

2. In the introduction also, in the second paragraph, the authors discuss about global initiatives to reduce C-section rates. But I missed the article that have been published recently in BMJ open quality describing a successful initiative in a public Brazilian hospital to reduce the local C-section rate (Negrini R, Ferreira RDdS, Albino RS, et al. Reducing caesarean rates in a public maternity hospital by implementing a plan of action: a quality improvement report. *BMJ Open Quality* 2020;9:e000791. doi:10.1136/bmjopen-2019-000791).

Our response

Thanks for this suggestion. We included the suggested paper by Negrini et al which is certainly an encouraging intervention to be implemented in the region.

The text now reads

“Along with clinical interventions, non-clinical interventions such as antenatal education, continuous emotional support during labour, audit and feedback, and the implementation of clinical guidelines have shown modest but statistically significant reductions in CS rates (7,16,17). A quality improvement study conducted in Brazil presented encouraging results. The project demonstrated a reduction in CS rates through a multilevel intervention targeting healthcare professionals. The interventions included reinforcing the use of analgesia during birth, continuous professional education and feedback over CS rates to the teams, among other measures (18). The World Health Organization (WHO) also recommends implementing non-clinical interventions to reduce unnecessary surgical births and it encourages the incorporation of an initial formative research process to tailor interventions to local contexts, ensure implementation feasibility, and anticipate specific barriers or identify facilitators (19,20)”.

The authors cited, in the introduction also, that the c-section rate has been increasing in Argentina but they did not bring the temporal evolution of this rate. I think is important to the readers to have this information.

Our response

We agree with your suggestion. We have included a paragraph on how the rates evolved in the public sector in Argentina as follows.

The text now reads

“Argentina is a federal country with a two-tier health system: a public system financed by government funds and a private system mix of social insurance and private insurance. Virtually all births occur at healthcare institutions (99.3%), and 64% of them within the public health system. Specialists lead obstetric services in Obstetrics and Gynaecology (OB/GYN) while the role of midwives depends on

how the hospital organises its services and human resources. Midwives provide antenatal care and depending on the institution, can assist or lead low-risk vaginal births”.

3. Why the C-section rate of 27% was chosen by the authors as cut-off for choosing the study hospitals?

Our response

Thank you for your comment on the threshold of 27% for the CS rate which was reported in the manuscript as eligibility criteria for the hospitals. We have deleted this criterion because it was an error in the writing of the manuscript. All hospitals included in the study have a CS rate higher than 27% (ranged from 27% to 50%). In the initial discussions with the Directors of the hospitals, it was required that the CS rate was high enough to allow for recognition of the possible or probable overuse, but this was not made explicit in quantitative terms.

The text now reads

“The eligibility criteria for hospitals to participate in the study were: a) public hospitals with ≥ 1000 annual childbirths, and b) use of the National Perinatal Information System (25,26). Out of the 88 potentially eligible institutions, 24 were non-randomly selected to represent the six Argentinian regions. Nineteen agreed to participate and obtained approval from local Ethics Committees (Figure 1), and five institutions declined to participate. Finally, five Argentinian regions were included in this study: Centre, North, Northeast and Buenos Aires City. In 2016, the median CS rate in the 24 eligible hospitals was 37%, ranging between 27% and 52% (2)”.

4. It's clear that hospitals of all Argentina's regions was chosen, but I think is important to specify how many hospital from each region.

Our Response

As suggested by the reviewer, we have included regions and the number of selected hospitals per each of them

The text now reads

“Women giving birth in the 19 participant hospitals represented 19% of the total live births in Argentina's public sector in 2017 (Figure 1). The hospitals corresponded to the North region (four hospitals), the Northeast (two hospitals), the centre (seven hospitals), and capital city (six hospitals). Five hospitals (three from the West-Cuyo Region and two from the South) either declined to participate or did not complete the endorsement by their ethics committee”.

5. the absent of Pain relief is a known issue the lead pregnant to choose the c-section as a delivery mode. I think it could be more discussed. It's interesting to bring the pregnant perspective raised in other articles, for example, Stoll K, Hauck YL, Downe S te al. Preference for cesarean section in young nulligravid women in eight OECD countries and implications for reproductive health education. *Reprod Health*. 2017 Sep 12;14(1):116. doi: 10.1186/s12978-017-0354-x.

Thanks for your suggestion. We further discussed the issue of pain relief -including new data from our formative research study- as well as the suggested reference.

The text now reads

“Our research suggests that services are organized to provide adequate responses to obstetric emergencies. However, they informed deficits in the availability of strategies for pain management such as epidural, hot showers or relaxation techniques, and limited antenatal education, all of which are important issues for women (35). Some hospitals had insufficient human resources and space limitations impeding constant support during birth. Evidence worldwide suggests fear of pain or lack of pain relief during labour as one of the primary factors underlying women preference for CS (7,15,36,37). Our formative research further reinforces the importance for women to have access to pain management strategies. The findings indicate that women from these hospitals prefer vaginal birth over CS, but CS preference is influenced by the pain experienced during birth (38).

The interviews with key informants acknowledged restrictive access to pain relief and its impact on the birth experience, particularly in adolescents. The management of pain during birth is an essential

component of obstetric services and needs to be organised, resourced, and monitored. This is particularly true in Argentina, which has a 13% overall adolescent pregnancy rate and provinces with rates above 20% (2). As well, a recurring topic that appeared when conducting semi-structured interviews was the need to improve antenatal education, which has already been identified as a non-clinical intervention with a positive impact on birth experience and vaginal birth rates (7,35,39)".

6. Most of the interviewee believe that clinical conditions are responsible for the increasing in the c-section rate. I guess it is a common excuse for the most of midwives and doctors in the obstetric area. Because of this, Robson created his classification. The authors cited that the knowledge of interviewees regarding Robson classification was part of the survey, but they did not mentioned the results of this question. It would be interesting to build a parallel among these information (Knowledge of Robson's classification x the belief that clinical conditions are responsible for high C-section rates).

Our Response

Thanks for your suggestion. In our preliminary analysis we did not find an association between the two variables (Knowledge of the Robson Classification and the belief that clinical conditions are responsible for high C-sections rates). A possible explanation is that the study was not designed to explore that association. We added one sentence to further explain this issue to the reader.

The text now reads

"Finally, six out of ten healthcare professionals agreed that there was a need to optimise CS use and half indicated they were familiar with the Robson Classification with no differences in terms of their opinion on the influence of clinical caseload and the knowledge of the Robson classification (data not shown)".

Discussion

[..] "In addition, the WHO recommends the use of the Robson classification to assess, monitor and compare CS rates in a standardized and action-oriented manner (19,30). This classification helps to understand the obstetric case mixed of a maternity and thus can challenge some common myths about causes of increasing CS rate such as obstetrical condition conducive to CS"

Reviewer 2

Reviewer: Dr. Helena Litorp, Uppsala Univ

Comments to the Author:

Thank you for asking me to review this interesting paper on determinants of CS (over)use in Argentina. I think the paper adds further understanding and knowledge to the increase in CS rates observed globally and specifically in Latin America and with results in line with other reports on the subject. Please find some specific comments below.

Introduction

* The introduction is well-structured and interesting, in particular I like the description of formative research and that the authors argue for why they chose this method for the current study.

1. I would, however, recommend to also state the national CS % in Argentina, preferably both in total, in private and in public sector (that is on the population level).

Our response

Thanks for your positive comments and constructive comments. As suggested, we have included a more detailed data on the progression of CS rates in Argentina. Unfortunately, official data from the private sector is lacking.

The text now reads

"Argentina, a middle-income country, reported rates between 27% and 52% within the public sector in 2017, while official data from the private sector is unavailable (2). According to reports by the Perinatal Reporting System, from 2009 to 2017, the use of CS has increased in the public sector by

22%, from 28% to 34%, with striking rates in some provinces being close to 50% (2). Similarly, Brazil registered national rates of 55%, with private providers close to 90% (3)".

2. I would recommend to rephrase the sentence of the aim of the study to include more of the study objectives (which I understand also included a survey on EmOC services available).

Our response

As suggested, the aim and objectives of the formative research were included.

The text now reads

"We conducted formative research to enrich understanding of the reasons for the sustained increase in CS use in Argentina's public sector with the aim to subsequently use this knowledge to tailor interventions in Argentina (21). We applied a mixed-methods approach to explore multiple layers and perspectives of both women and health providers, and to assess the characteristics of the obstetric services in the participating hospitals including the availability of and access to pain management interventions (e.g. anaesthesia and non-pharmacological interventions), and the availability of and access to emergency obstetric services. Formative research has its origins in ethnographic studies and uses various data-gathering processes. The rationale behind this approach is to consider the peculiarities of the setting to better tailor suitable interventions and, at the same time, to catalyse the target population into change (22–24).

This article will focus on the perspectives of specialists in OB/GYN, midwives and OB/GYN residents (trainees) on determinants of CS use in Argentina's public maternity hospitals".

3. Methods:

Pls motivate why only public hospitals were included and why hospitals needed to have a CS rate above 27%.

Our response

Thank you for your comment on the threshold of 27% for the CS rate which was reported in the manuscript as eligibility criteria for the hospitals. We have deleted this criterion because it was an error in the writing of the manuscript. All hospitals included in the study have a CS rate higher than 27% (ranged from 27% to 50%). In the initial discussions with the Directors of the hospitals, it was required that the CS rate was high enough to allow for recognition of the possible or probable overuse, but this was not made explicit in quantitative terms.

The text now reads

"The eligibility criteria for hospitals to participate in the study were: a) public hospitals with ≥ 1000 annual childbirths, and b) use of the National Perinatal Information System (25,26). Out of the 88 potentially eligible institutions, 24 were non-randomly selected to represent the six Argentinian regions. Nineteen agreed to participate and obtained approval from local Ethics Committees (Figure 1), and five institutions declined to participate. Finally, five Argentinian regions were included in this study: Centre, North, Northeast and Buenos Aires City. In 2016, the median CS rate in the 24 eligible hospitals was 37%, ranging between 27% and 52% (2)".

4. The Discussion is overall well-balanced, but I think the section on obstetric care providers' view that the CS rate is caused by a complex case-mix would benefit from a discussion whether such arguments are warranted - can one really argue that around 40% of deliveries are so complex that they require CS given that complications demanding CS are in fact relatively rare events? Or would this rather be a way to withdraw from responsibility? Pls elaborate.

Our response

We agree with the reviewer in that the argument of complex case-mix does not sustain with CS rates of 40% or near 40% as in most of the Hospitals in this sample. We have elaborated more in the discussion on the use of the Robson classification to precise the nature of the case-mix as well as the use of tailored interventions targeting clinicians.

The text now reads

"All the professional groups agreed that CS is related to complex-case mix or clinical indications. This finding is in line with results from other international studies (11). Panda et al. summarised clinicians'

views of factors influencing decision-making in a systematic review of 34 studies published in English, and clinical indication appeared significant in eighteen studies (11). Still, the reported CS rates by most of the hospitals in this sample - median of 37% - suggest that case-mix alone cannot explain the high rates. Our online survey did not explore what providers understood as a clinical indication, but the semi-structured interviews further confirmed clinicians' views that increased comorbidities and new technologies contribute to the increased use of CS. These findings suggest that overall, healthcare teams perceived clinical indications as the main driver for performing a CS though about a third identified other contextual factors as determinants of CS use like services organisation, service provision, limited human resources, midwives' role, women's preferences, adequate number of professionals to support women during labour, or deficits in antenatal education for both pregnant women and companions. Though factors such an increase in the age of women, in the proportion of nulliparous women, in the prevalence of obesity, or in multiple births (in vitro fertilization) have been suggested as culprits of the increasing trend of the use of CS, these factors alone cannot explain the phenomenon. The WHO emphasizes that non-clinical factors such as convenience of a CS, fear of litigation and other organizational issues need to be recognized and addressed for successful interventions (39).

In addition, the WHO recommends the use of the Robson classification to assess, monitor and compare CS rates in a standardized and action-oriented manner (19,30). This classification helps to understand the obstetric case mixed of a maternity and thus can challenge some common myths about causes of increasing CS rate such as obstetrical condition conducive to CS".

Once again, we appreciate the opportunity of publishing in your Journal. Please do not hesitate to contact us if further clarifications are needed.

VERSION 2 – REVIEW

REVIEWER	Negrini, Romulo Santa Casa São Paulo - Brazil, Obstetrics and Gynecology
REVIEW RETURNED	20-Dec-2021

GENERAL COMMENTS	All questions asked were answered completely and satisfactorily.
--

REVIEWER	Litorp, Helena Uppsala Univ
REVIEW RETURNED	08-Dec-2021

GENERAL COMMENTS	Thank you, all queries I raised have been addressed in the revised version of the manuscript.
---